# The Role of Social Support in Participation Perspectives of Caregivers of Children with Intellectual Disabilities in India and South Africa

**DOI:** 10.3390/ijerph17186644

**Published:** 2020-09-11

**Authors:** Shakila Dada, Kirsty Bastable, Santoshi Halder

**Affiliations:** 1Centre for Augmentative and Alternative Communication, Humanities Faculty, University of Pretoria, Pretoria 0002, South Africa; kgb0071978@gmail.com; 2Department of Education, University of Calcutta, Alipore Campus, 1 Reformatory St., Kolkata 700027, India; santoshi_halder@yahoo.com

**Keywords:** social support, family support survey, participation, intellectual disabilities, low- and middle-income country

## Abstract

Caregivers are an intrinsic component of the environment of children with intellectual disabilities. However, caregivers’ capacity to support children’s participation may be linked to the social support that they, as caregivers, receive. Social support may increase participation, educational, psychological, medical and financial opportunities. However, there is a lack of information on social support in middle-income countries. The current study described and compared the social support of caregivers of children with intellectual disabilities by using the Family Support Survey (FSS) in India and South Africa. The different types of social support were subsequently considered in relation to participation, using the Children’s Assessment of Participation and Enjoyment (CAPE). One hundred caregiver–child dyads from India and 123 from South Africa participated in this study. The data were analysed using non-parametric measures. Indian caregivers reported greater availability of more helpful support than did the South African caregivers. Social support was associated with children’s participation diversity (India) and intensity (South Africa). The child-/caregiver-reported participation data showed different associations with participation. Results from this study suggest that perceived social support of caregivers differs between countries and is associated with their child’s participation. These factors need to be considered when generalising results from different countries.

## 1. Introduction

The introduction of the International Classification of Functioning, Disability and Health (ICF) [1] and the Child and Youth Version (ICF-CY) [2] highlighted participation as a critical health outcome [3]. Furthermore, participation has been highlighted as a human right for persons with disabilities at the United Nations Convention on the Rights of Persons with Disabilities [4]. Participation is described as an important means for achieving physical, social and academic development, cultural understanding, and community inclusion. It is argued that through participation, developmental skills are practiced until an outcome of learned skills is produced [5,6].

As the field of participation is growing, however, gaps in research have emerged. In spite of the fact that participation is reported to be influenced equally by personal and environmental factors [7,8,9], the bulk of research on participation has focused on personal factors. The current research has provided evidence of decreased participation for children with disabilities [9,10,11,12,13,14] and specific patterns of participation associated with the type and severity of a disability [15,16,17,18,19]. Studies considering the impact of environmental factors are more limited in number, but as highlighted by Anaby et al. in a scoping review on the effect of the environment on participation, family support and geographic location are facilitators of participation, while attitudes, the physical environment, policies and a lack of support serve as barriers to participation [20].

A paucity of research is also evident in relation to the effect of the income level of the country or culture on participation. Most children with disabilities in the world live in low- and middle-income countries [21], and environmental factors have been identified as an important participation-related concept. Hence, it would have been expected that research on participation in low- and middle-income countries would be common. However, the review by Anaby et al. [20], which identified 28 studies (and three reviews), all were conducted in high-income countries. A more recent scoping review by Schlebusch et al. [22] identified 78 studies on participation from low- and middle-income countries (55% conducted after the Anaby et al. [20] review). However, only 4% (*n* = 6) of these studies were from low-income countries, with 68% (*n* = 53) conducted in upper-middle-income countries. Furthermore, again only 4% (*n* = 6) of the studies in this review considered the effect of the environment on children’s participation, while the remaining studies investigated participation as a process (*n* = 7), participation as an outcome (*n* = 42), child-related outcomes (*n* = 14), and the measurement of participation or related constructs (*n* = 11) [22]. All in all, there remains a lack of research on participation of children with disabilities from low- and middle-income countries, particularly in relation to environmental factors [22,23].

The importance of research on participation from low- and middle-income countries relates specifically to differences in the environment that may affect children with disabilities’ participation. As indicated by Anaby et al. [20], environmental factors may function as facilitators of or barriers to the participation. Compared to their peers in high-income countries, children in low- and middle-income countries have been identified as being at greater risk from environmental influences such as poverty, reduced educational opportunities, violence and difficulty accessing healthcare [24,25,26]. In addition, the studies in the Anaby et al. review [20] were mostly from English-speaking countries embracing Eurocentric/western philosophies such as the U.S., the U.K., Canada, Australia and Europe [20], which see the individual as being independent from their community. This is in contrast to Afro-/Asia-centric philosophies that are founded on collectivism or see the self as inseparable from the community [27,28,29]. Differences in life philosophies may affect perceptions of self and disability, perceptions or availability of support, communication, and hence participation in these settings [27,28,29].

Within different cultural philosophies, the role of caregivers and the impact of factors such as caregiver support may affect participation. Unfortunately, limited research has been conducted in this area. Caregivers play a much greater role in finding [30,31] and facilitating [32] opportunities for participation for children with disabilities [20,33] than for children with typical development. In fact, the responsibility of ensuring that the rights of a child with intellectual disabilities are met, is reported to fall most often on caregivers [34,35,36]. Such responsibilities can create additional stress for caregivers and may limit their adaptability. Different forms of social support have been described as buffers for caregivers of children with disabilities to decrease stress and increase positive parenting [37,38]. When considering participation specifically, as expressed in the ICF-CY [2], “the role of the family environment and others in the immediate environment is integral to understanding participation, ….” [2], (p. xvi). Yet, little is known about the support experienced by caregivers of children [39], particularly in Afro-/Asia-centric countries [40]. Social support specifically is a process that “arises from formal support (medical or professional) and informal sources (extended family, friends, and neighbours) around the caregiver and family” [40], (p. 160). Social support is said to be a reciprocal interaction in which caregivers feel cared for, esteemed and valued, and in which they are engaged in a system of communication and mutual responsibility [41]. Social support enables caregivers of children with disabilities to mediate the stress that they face [38,42,43] by developing resilience [44] and increasing their situational appraisal [45] and coping strategies [46]. While reductions in stress are reported to increase well-being [47], the presence of social support for caregivers and the use of positive caregiving styles are reported to increase the quality of caregiving [48]. Nurturing the child’s self-esteem can also result in better developmental outcomes for the child [40,48].

One distinct difference between Eurocentric and Afro-/Asia-centric households is the proportion of multi-generational households (both India and South Africa) [40,49]. Multigenerational households have been highlighted as able to provide resilience and growth where this might otherwise not have been possible [50,51,52,53]. The presence of older generations in the household can, however, also add to a caregiver’s responsibilities. In Eurocentric cultures, help for an older generation is provided primarily when specific needs arise (for example injury or illness), and therefore multigenerational households are less common. In Afro-/Asia-centric cultures, simply “being old” is sufficient for the provision of additional support [51,52], and the provision of this support is culturally obligatory [52,53].

The influence of caregivers of children with disabilities on their child’s participation is represented in the context- and environment-related constructs of the family of Participation-Related Constructs (fPRC) model [6]. In this model, caregivers constitute a key component of their child’s context (part of the environment). They provide opportunities for participation, regulate the environment, and respond to their child [54]. In spite of this key role played by caregivers of children who have intellectual disabilities, only four studies [32] have made use of tools in which the caregivers reported on participation, and none of these considered factors specific to the caregiver which may affect participation [6]. From a systems perspective, the impact of caregiver factors on the participation of a child with intellectual disabilities can also be appreciated, as the influence of each level of the system on the other levels is highlighted. This perspective is supported by studies that identify the caregiver education level, income and social support structures [33,55] as factors that may have an effect on participation [33,56,57].

The final gap that has been noticed in the literature is the notion of diagnosis. The most commonly reported disability in terms of participation is cerebral palsy [20,58,59,60,61,62,63], and research on other conditions such as intellectual disabilities is sparse. Intellectual disability is a pervasive and lifelong condition in which children present significant limitations with regard to intellectual functioning and adaptive behaviour, prior to the age of 18 years [64]. In addition to the individual challenges experienced by children who have intellectual disabilities, environmental barriers may impede the achievement of human rights [34,35,36]. These may include a lack of opportunities for participation in education, recreation, leisure, sporting and community activities [10,65]. For children with intellectual disabilities, the combination of individual challenges and environmental barriers can result in decreased cognitive and linguistic skills, poor motor and social skills [66], social isolation and chronic health problems [32]. A systematic review of the participation of children with intellectual disabilities identified four studies that reported that children with intellectual disabilities participated to a similar extent in leisure activities, but less in social activities within the community, recreational activities, family enrichment activities and formal activities, than did their typically developing peers [32]. Other studies not included in the review indicated decreased participation in active-physical and skills-based activities [19,66,67] and a higher proportion of participation in social and recreational activities [19]. In addition, children with intellectual disabilities were noted to participate in a significantly greater number of activities at home [68], by themselves [19] or with adults, rather than with peers [10], in comparison to children with typical development. In addition, challenges in the participation of a child with an intellectual disability were found to affect not only the child, but also to place high levels of stress on the parents and family [30].

In conclusion, there is a need to describe the influence that the environmental component of caregiver support has on the participation of children with intellectual disabilities from low- and middle-income countries [33]. The current study aimed to measure, describe and compare the social support of caregivers of children with intellectual disabilities from India and South Africa, and to determine if there is an association between the social support reported by caregivers and the participation of their children as reported by caregivers and their children. India and South Africa were selected since both countries have been identified as having cultures in which households are more commonly multigenerational. However, India is a lower-middle-income country and has a very high reported prevalence of intellectual disability (≈6%), while South Africa is an upper-middle-income country and has a lower reported prevalence (≈2.25%) of intellectual disability [69,70].

## 2. Aims

This study had three key aims: firstly, to describe and compare the social support reported by caregivers of children with intellectual disabilities in India and South Africa; secondly, to determine if there was any association between the demographic factors and the social support reported by caregivers; thirdly, to determine if there was any association between the social support reported by caregivers and the participation of their children with intellectual disabilities.

The first hypothesis formulated for this study was that the social support available to caregivers in India and South Africa would be different. The second hypothesis suggested that demographic factors in India and South Africa would affect social support, and the third hypothesis stated that increased perceived social support would be associated with increased participation of the children with intellectual disabilities.

## 3. Materials and Methods

### 3.1. Study Design, Sampling and Participant Selection

A comparative group design was used for this study. Purposive sampling was used in schools for children with intellectual disabilities to identify participants. In both countries, education for children with disabilities is provided in special schools which can be either government funded or private. The support provided by these schools is dependent on a range of factors including context (rural/urban), funding and fees paid by parents. Both urban and rural schools were included in this study.

Inclusion criteria for caregiver–child dyads required children to be between the ages of 6 and 21, and to have a primary diagnosis of mild to moderate intellectual disability. Caregivers were required to be literate in Bengali, English, Afrikaans, isiZulu or isiXhosa, and children were required to speak Bengali, English, Afrikaans, isiZulu or isiXhosa. If a child’s home language was not the same as the language in which the Children’s Assessment of Participation and Enjoyment (CAPE) [71] was to be administered at their school, then the child needed to have been schooled in the language of the CAPE [71] for at least 1½ years in order to be included in the study.

### 3.2. Ethics

Ethics approval for the study was obtained from the relevant ethics committees of the institutions of higher education in both countries. Permission was obtained from the appropriate departments and heads of schools or centres. In India, participants were identified in twelve schools and centres for children with disabilities. In South Africa, permission was obtained from the Department of Education in six provinces. Permission was also obtained from the principals and governing bodies of the schools identified. Eleven government schools and four private schools gave permission for their children to participate in the study.

### 3.3. Participants

A total of 223 caregiver–child dyads participated in the study, with 100 dyads from India and 123 from South Africa. The children had a mean age of 12:4 (years:months), and the sex composition of the sample was 61.3% male and 38.7% female. Although the reporting caregiver was primarily the child’s mother (73.6%), fathers (15.6%) and other caregivers (10.8%) also reported on their children. More than half of the caregivers had at most a Grade 12 education (India 68%; South Africa 64%), and 64% of caregivers reported a household income of less than ZAR 30,000.00 (approximately EUR 1500.00) per month. Indian caregivers reported between one and six children residing in the household (M = 2), while the South African caregivers reported between one and 13 children (M = 3) in the household. Indian caregivers reported having grandparents living in the house in 67.4% of families, and other family members in 47.7% of families. South African caregivers reported having grandparents living in the house in 32.6% of families, and other family members in 52.3% of families. Statistically significant differences were evident in the demographic data of caregivers from India and South Africa.

The summarised demographic data of the participants are presented in Table 1.

### 3.4. Materials

The availability of support to caregivers of children with intellectual disabilities in this study was determined using the Family Support Scale (FSS) [72]. The FSS [72] is a 19-component scale that asks caregivers to rate the helpfulness of support from various sources, for example, spouse, parents, friends, and parent groups. For each support source, the caregiver indicated on a Likert scale whether the support was not available (0), not at all helpful (1), sometimes helpful (2), generally helpful (3), very helpful (4) or extremely helpful (5). In the scoring of the FSS [72], the 19 sources were grouped into four factors—namely, family, spousal, social and professional support [73]. The FSS [72] was highlighted as a measure suitable for use with caregivers of children with disabilities in a scoping review on the subject [40], and was reported to have both internal consistency and stability across samples [73].

The participation of children was reported using the Children’s Assessment of Participation and Enjoyment (CAPE) [71]. The CAPE [71] is a self-report questionnaire that has been developed for use with children/youth between the ages of 6 and 21 years, with and without disabilities. The CAPE [71] considers 55 activities grouped into domains (overall, informal and formal) or activity types (recreational, active-physical, social, skills-based and self-improvement). For each activity, five dimensions of participation are measured—namely, diversity, intensity, companionship, location and enjoyment [71]. A proxy report of the CAPE [71] was used to measure the caregivers’ perceptions of their children’s participation [74]. As reported in Dada, Bastable, Schlebusch and Halder [74] and available in this special edition of the Int. J. Eviron. Res. Public Health, the internal consistency of the CAPE [71] for this study was excellent (0.923 < α < 0.993) [74,75].

All materials for this study were translated into Afrikaans, Bengali, Sepedi, isiXhosa, and isiZulu. Translation included forward and blind backward translation, as well as the consideration of linguistic, functional and cultural equivalence [76].

### 3.5. Data Collection

A total of 422 information packs were sent to caregivers via their child’s school. The return rate in India was approximately 70% and South Africa approximately 55%.

The information pack included information on the study, a written consent form, the FSS [72], and the proxy version of the CAPE [71] in the language of teaching and learning at the child’s school. The consenting caregivers completed these forms and returned them to the school in an envelope. The children whose caregivers consented were asked to provide assent and they completed the CAPE [71] in an interview at their school. All children in India assented and 98% of South African children assented. The CAPE [71] interview was conducted in close adherence to the instructions and using the visual supports provided in the manual. The interview was conducted in the language of teaching and learning at the school or the child’s home language, with the researcher reading the questions to the child and recording their answers on the CAPE [71] forms. Children who assented, as well as those who did not, were provided with a token of appreciation (a ruler and an eraser).

### 3.6. Data Analysis

Data analysis for this study was conducted using SPSS version 26 (IBM, Armonk, NY, USA) [77]. Demographic data, data from the FSS [72] and from the CAPE [71] were analysed for normality first, and then using non-parametric tests including Pearson’s chi-square, Fisher’s exact test, Mann–Whitney U, Kruskal–Wallace, and the independent samples test. Internal consistency of the FSS [72] was evaluated using Cronbach’s alpha [75]. Due to significant differences being evident in the demographic data of participants from India and South Africa, the analysis of the CAPE [71] and associations between the CAPE [71] and FSS [72] were conducted on each set of data independently, rather than as a single set.

The participation data from India and South Africa were compared to their respective FSS [72] data using Kendall’s Tau_b_ [78] to determine association. Although both Pearson’s and Spearman’s correlation coefficients are better known statistical coefficients, Kendall’s Tau_b_ has been shown to be less sensitive to outliers, thereby limiting the number of Type 1 errors and providing tighter confidence intervals and clearer interpretation [78,79]—specifically where sample sizes are smaller [80].

## 4. Results

The internal consistency of the FSS [72] is reported on first, followed by the social support perceived by caregivers. This is followed by the associations in demographic and FSS [72] data. The participation data are summarised (the full data are available in the paper titled: The participation of children with intellectual disabilities: Including the voices of children and their caregivers in India and South Africa, in this special edition), and associations between social support and participation are presented.

### 4.1. Internal Consistency of the FSS [72]

The internal consistency of the FSS [72] was determined using Cronbach’s alpha. The FSS [72] presented with Cronbach’s alpha coefficients between 0.748 and 0.780, which are considered acceptable [75,81].

### 4.2. Social Support Reported by Caregivers of Children with Intellectual Disabilities in India and South Africa

On average, caregivers in India and South Africa reported that family and spouse groups were generally helpful (family mean = 2.72, spouse mean = 2.70), and social and professional groups were sometimes helpful (social mean = 1.67, professional mean = 2.23). The caregiver’s parents and spouse were most likely to be reported as extremely helpful. Parent groups, co-workers, social groups, church or spiritual support, early childhood intervention centres, and governmental and non-governmental agencies were most likely to be unavailable to caregivers. No significant differences were evident between social support factors for India and South Africa, except for spousal support (*p* = 0.000). For specific support sources, significant differences were evident between India and South Africa, both for the level of support reported and the sources available. Overall, the caregivers in India reported greater helpfulness from available support sources, but older children, co-workers or parent groups, social and religious groups, early childhood intervention and governmental/non-governmental services were not available to the majority of families. The South African caregivers, however, reported that social support groups were less helpful to them. Unavailable support in South Africa included relatives, spousal friends, friends, neighbours, other parents, parent and social groups, early childhood intervention and governmental/non-governmental services. The full social support results are indicated in Table 2.

#### Associations between Demographic Factors and Social Support

In India, the association indicated increased family support when the caregiver was the mother (Tau_b_ = 0.194_)_, whereas in South Africa, decreased family support was indicated when the caregiver was the mother (Tau_b_ = −0.163). Small positive associations between social support (Tau_b_ = 0.201) and employment (Tau_b_ = 0.157) were evident in India, while a small effect of home language on professional support was indicated for South Africa (*p* = 0.262). Associations between child sex and age were seen in India, with increased support for male (Tau_b_ = −0.182) and younger children (Tau_b_ = 0.173) [79]. The association data are presented in Table 3 below.

### 4.3. Participation and Social Support for Children with Intellectual Disabilities in India and South Africa

#### 4.3.1. Self-Reported Participation of Children with Intellectual Disabilities

Children in India and South Africa participated in a similar number of activities overall and with the same enjoyment. However, children from India were noted to participate more frequently at home with close family, while the children from South Africa participated less frequently at a relative’s house with extended family (medium to large effects). The full participation data are available in Dada, Bastable, Schlebusch and Halder, in this special edition of the Int. J. Eviron. Res. Public Health.

#### 4.3.2. Proxy-Reported Participation of Children with Intellectual Disabilities

Caregiver-reported participation differed from self-reported participation across both India and South Africa in terms of the number of social and skills-based activities participated in and to the level of enjoyment. No significant differences in the reporting of intensity, with whom, or where activities were conducted were evident. The full caregiver participation results are available in the Dada, Bastable, Schlebusch and Halder, in this special edition of the Int. J. Eviron. Res. Public Health.

### 4.4. Association between Social Support and Participation

#### 4.4.1. Association between Social Support and Caregiver-Reported Participation of Children with Intellectual Disabilities in India and South Africa

Associations with the presence of family support sources were evident for the intensity of participation overall, in the informal domain and for self-improvement activities in South Africa. Family support was also associated with where formal and skills-based activities occurred in South Africa, but with caregiver-reported enjoyment in active-physical activities in India. Spousal factors were associated with the diversity of social activities in South Africa, with whom and where participation occurred overall, and with participation in the informal domain. In India, spousal support was associated with whom recreational activities occurred, and where informal and social activities took place. The social support factor was associated with the diversity of participation overall, in both the informal and formal domains in India. In South Africa, however, social support was associated with the intensity of participation overall, participation in the informal domain, and social activities. Intensity of participation in the formal domain as well as where participation in this domain occurred was associated with social support in India. Professional support was associated in India with the intensity of activities in the formal domain and social activities, but in South Africa, it was associated with participation with whom and participation in the informal domain. The association data are presented in Table 4.

#### 4.4.2. Associations between Social Support and Child-Reported Participation of Children with Intellectual Disabilities in India and South Africa

Children in South Africa indicated associations between participation and family and social support, while children in India indicated connections between active physical activities and family. Intensity of participation in social (India) and recreational (South Africa) activities was associated with social support. An association between with whom recreational activities occurred and family support was evident in South Africa, while family support was associated with where participation occurred in India as well as South Africa. All effects identified were small [75]. Significant associations are reported in Table 5.

## 5. Discussion

Intellectual disability is one of the leading developmental disabilities in low- and middle-income countries [69,70]. For caregivers, a child with an intellectual disability can increase the stress and demands of parenting. Caregivers may find themselves solely responsible for ensuring that their child’s rights are recognised and their needs are met [34,35,36]. Yet, in the face of increased demands, caregivers have reported a lack of support from outside of their immediate family [82,83]. Increased stress for caregivers can limit their ability to support their children and provide them with the required developmental opportunities [37,38] through participation in activities [6]. The relationship between the caregiver and the child’s participation has been described as a related factor that may influence participation (the environment) [7,8,9,84]. For caregivers, however, social support has been described as a buffer to stress [37,38], which may increase their capacity to facilitate their child’s participation. This study sought to describe the different types of social support experienced by caregivers of children with intellectual disabilities in India and South Africa (middle-income countries) and to identify whether there is an association between the social support reported by caregivers and the children’s participation.

As discussed previously, the bulk of research on participation originated in high-income countries [22]. Hence, this study sought to provide information on participation of children with intellectual disabilities in two middle-income countries. In the initial analysis of demographic information from the participants it became clear that although both India and South Africa are middle-income countries, significant differences were evident in the demographics of the caregiver groups from these two countries. Education, income and employment differed significantly among the caregivers, with the South African caregivers reporting lower levels of education, income and employment. Such differences highlight the need for research across both low- and middle-income countries, as demographic differences alone make generalisation from one country to another challenging.

The presence of multigenerational households has been hypothesised to affect social support structures and to be widely prevalent in collectivist cultures. Nonetheless, only half of the families in this study came from multigenerational households, with Indian caregivers reporting significantly more multigenerational households than caregivers in South Africa—despite the fact that it has been suggested that multigenerational households may provide additional support for caregivers of children with disabilities [50,51,52]. Our study suggests that it cannot be assumed that households from traditionally collectivist countries will contain multiple generations, even if this has been a cultural norm in the past. Nowadays, industrialisation and globalisation have a clear impact on societal functioning [53].

In spite of the demographic differences identified between India and South Africa, social support from family, social and professional factors was reported as similar, but spousal support was significantly different. Caregivers from India reported more support from spousal sources—including their spouse, spouse’s parents—and friends than did caregivers from South Africa, while approximately a quarter of South African caregivers reported that their spouse and relatives were not available. The lack of spousal availability in South Africa may result from the country’s past, as the systematisation of migrant labour under apartheid split families by allowing only the working individual to stay in an urban area. As a result, families were divided, with mothers and children living in a different place from fathers. The forced separation of families under apartheid has had a significant effect on family structure in South Africa, which is still experienced today [85]. In Indian culture (in contrast), once married, some women would traditionally live with their husband’s family, hence having a spouse available may also include the support of his family [53].

The demographic factors of relationship to child (family), employment (India, social), home language (South Africa, professional), child sex (India, family) and child age (India, social) were associated with social support reported by caregivers, although the associations showed small effects. The support experienced by caregivers may well be related to the social structure of the country, including how neighbours and friends support working parents, and cultural biases relating to sex and age [86,87,88]. For example, in India, male children are often revered while female children may be seen as a burden [89,90], while in South Africa professional support is most often available to caregivers in English or Afrikaans, which may not be their home language [91].

Overall participation of children with intellectual disabilities in India and South Africa was similar, but differences were evident in the formal domain, as well as in respect of active-physical and recreational activities. Weak positive associations with social support were evident across both the Indian and South African data in relation to the diversity of participation (mostly family support, but also spousal and social support). As participation in activities for children with intellectual disabilities requires (in many situations) the availability of the activities, as well as a partner to facilitate the child, the presence of additional family support may reduce the load on the caregiver and provide more opportunities for the child to participate. Similarly, spousal and social support may increase the number of opportunities for the child to participate.

The effect that social support given to caregivers of children with intellectual disabilities has on the participation of their child is evident in the associations identified between support sources and the caregiver-reported participation. In India, associations were seen most often between social and professional support and the formal domain, while in South Africa associations between spousal and social support were evident more often in the informal domain. These differences could be linked to the availability of resources. With lower income reported by caregivers in South Africa, it is possible that informal activities place less of a financial burden on caregivers. Importantly, however, the South African caregivers reported more households where the spouse was not available than did the Indian households. Thus, the association between both spousal and social support is logical, as when spousal support was not available, the South African caregivers relied on extended family and friends for support.

It is interesting to note the differences in associations evident between participation data as reported by the children and caregivers—this is in spite of the data not being significantly different [74]. The children’s participation data were associated with family and social support, mostly in informal activities for South Africa, but primarily in formalised activities for India. Enjoyment was associated with professional support for active-physical activities in South Africa. This is a logical conclusion, in that children with special needs may require devices or support to participate in active-physical activities. Such support is often provided by professionals or professional organisations, for examples the special Olympics. Of interest, however, is that the enjoyment of formal (India) and self-improvement (South Africa) activities was associated with family support for children. In this regard it may be that self-improvement activities are participated in most frequently at home with family, or that these activities are most important to the family—hence all family members contribute towards the child’s enjoyment in these areas.

Overall, the data from our study provide evidence that environmental factors play a role in the social support that caregivers of children with intellectual disabilities receive. Sequentially, social support plays a role in the participation of children with intellectual disabilities. Differences in the associations between social support and caregiver-reported participation point to demographic and cultural influences on the participation of children with intellectual disabilities. At the same time, differences in social support and participation associations between the child- and caregiver-reported participation data emphasise the subjective nature of social support and participation. Hence, results should not be generalised from one country to the next, even when aspects of their cultures appear similar at face value. When considering the participation of children with intellectual disabilities, the family environment should be examined as a whole, with reporting from multiple members in order to understand the factors that affect participation.

Considered in relation to current models of participation, the effects of social support were mostly weak, yet consistent across multiple areas of participation. Hence, they cannot be ignored in the consideration of participation of children with intellectual disabilities. Although current models of participation such as the fPRC [84] now include the child’s context and environment, they have until recently focused more on the direct associations between the child and the environment.

### 5.1. Recommendations

Recommendations arising from this study include the exploration of the role of environmental factors in the participation of children with intellectual disabilities in other countries. Specifically, further research is recommended on the effects that social support interventions have on the participation of children who are typically developing and those with disabilities.

### 5.2. Limitations

The FSS [72] used in this study focuses on the perceived helpfulness of different types of social support but does not provide an opportunity for caregivers to report on the context of the support or to identify alternative supports that are needed [40]. Perhaps additional measures of the type of social support that is needed could have been included.

## 6. Conclusions

The social support provided to caregivers of children with intellectual disabilities in India and South Africa was similar in many respects. However, social support is sensitive to demographic factors such as employment and the relationship of the caregiver to the child. Caregivers of children with intellectual disabilities overwhelmingly reported a lack of social and professional support. In both India and South Africa, studies showed positive associations between participation and social support. For India, increased social support was associated with increased diversity of participation, while in South Africa it was associated with increased intensity of participation. Differences in results from different countries may preclude the generalisation of results relating to both social support and participation.

## Figures and Tables

**Table 1 ijerph-17-06644-t001:** Demographic data of participants.

Demographic Factor	India (*n* = 100)	South Africa (*n* = 123)	Combined Data (*n* = 223)	Equivalence *p*-Value
Caregiver respondent (%) ^1^				
Mother	33.8	39.8	73.6	0.129 ^3^
Father	6.1	9.5	15.6	
Other	3.5	7.4	10.8	
Caregiver education (%) ^1^				
Grade 11 or less	23.1	17.8	40.9	0.000 *^,2^
Grade 12	7.1	17.8	24.9	
Degree	12.0	8.9	20.9	
Other	2.2	11.1	13.3	
Household income per month (%) ^1^				
<ZAR 4500 (≈EUR 220)	1.8	26.2	28.0	0.000 *^,2^
ZAR 4501–ZAR 12,500 (≈EUR 600)	4.4	12.4	16.9	
ZAR 12,501–ZAR 30,000 (≈EUR 1500)	11.6	7.6	19.1	
ZAR 30,001–ZAR 52,000 (≈EUR 2500)	5.8	3.6	9.3	
ZAR 52,001–ZAR 70,000 (≈EUR 3370)	6.2	3.1	9.3	
>ZAR 70,001 (≈EUR 3370)	14.7	2.7	17.3	
Number of children in the household (Median)	2.0	3.0	2.0	0.000 *^,2^
Grandparents in household (%) ^1^	67.4	32.6	46.6	0.000 *^,2^
Additional family members in household (%) ^1^	47.7	52.3	50.6	0.000 *^,2^
Employment (%) ^1^				
Home executive/housewife	5.1%	20.8	14	0.000 *^,2^
Not working currently	1.0	15.4	9.2	
Working part time	11.1	12.3	11.8	
Working full time	82.8	43.1	60.3	
Other	0.0	8.5	4.8	
Child younger than 13 years (%) ^1^	51.0	43.1	46.5	0.232 ^2^
Child 13 years or older (%) ^1^	49.0	56.9	53.5	
Sex (%) ^1^				
Male	66.0	57.8	61.3	0.440 **^3^**
Female	34.0	42.2	38.7	

Note * the p value is significant. ^1^ Percentages may not add up to 100 due to rounding. ^2^ Pearson’s chi-square *p* < 0.05. ^3^ Fisher’s exact test–one-sided.

**Table 2 ijerph-17-06644-t002:** Caregiver-reported social support using the Family Support Survey (FSS) [72].

Family Support Scale ^1^	India (*n* = 100)		South Africa (*n* = 123)		*p*-Value ^2^/ Mann-Whitney U ^2^
	Mode	% Not Available	Mode	% Not Available	
**Family (Mean)**	2.69	1.00	2.74	4.80	0.931 ^3^
My parents	4.00	18.0	5.00	12.40	0.000 *
My relatives	4.00	5.00	0.00	33.90	0.000 *
My older children	0.00	70.00	0.00	19.30	0.000 *
**Spousal (Mean)**	3.23	3.00	2.23	6.50	0.000 *^,^^3^
My spouse’s parents	4.00	25.00	0.00	30.60	0.000 *
My spouse’s relatives	4.00	7.00	5.00	23.60	0.000 *
My spouse	5.00	7.00	0.00	24.50	0.000 *
My spouse’s friends	4.00	13.00	0.00	42.50	0.000 *
**Social (Mean)**	1.54	2.00	1.78	14.60	0.230 ^3^
My friends	4.00	13.00	0.00	35.10	0.000 *
My neighbours	4.00	8.00	0.00	36.00	0.000 *
Other parents	4.00	10.00	0.00	43.20	0.000 *
Co-workers	0.00	66.00	0.00	48.10	0.000 *
Parent group members	0.00	78.00	0.00	68.60	0.118
Social groups	0.00	89.00	0.00	59.40	0.000 *
Church/spiritual	0.00	96.00	0.00	40.40	0.000 *
**Professional (Mean)**	2.12	6.00	2.33	4.00	0.431 ^3^
Family doctor	4.00	9.00	0.00	31.90	0.000 *
Early childhood intervention	0.00	91.00	0.00	52.30	0.000 *
School/day care	4.00	19.00	0.00	36.30	0.000 *
Professionals	4.00	16.00	5.00	27.50	0.000 *
Organisations Non-/Governmental	0.00	92.00	0.00	67.30	0.000 *

^1^ Scores were measured on a Likert scale: 0 = not available; 1 = not at all helpful; 2 = sometimes helpful; 3 = generally helpful; 4 = very helpful; 5 = extremely helpful. ^2^ Pearson’s chi-square, *p* < 0.05. ^3^ Mann–Whitney U, * P is significant when *p* < 0.05.

**Table 3 ijerph-17-06644-t003:** Associations between demographic social support factors in India and South Africa.

Demographic Factors	Family		Spousal		Social		Professional	
	India	South Africa	India	South Africa	India	South Africa	India	South Africa
Relationship to child ^1^	0.023 *	0.030 *	0.957	0.067	0.221	0.807	0.150	0.215
Education ^1^	0.466	0.566	0.112	0.816	0.773	0.761	0.511	0.364
Employment ^1^	0.940	0.112	00.577	0.205	0.016 *	0.139	0.547	0.249
Income ^1^	0.716	0.315	0.041 *	0.946	0.858	0.013	0.540	0.262
Home language ^2^	0.806	0.420	0.906	0.234	0.560	0.059	0.166	0.031 *
Number of children in the home ^1^	0.218	0.246	0.711	0.311	0.290	0.814	0.290	0.287
Number of grandparents in the home ^1^	0.693	0.214	0.136	0.350	0.335	0.824	0.587	0.419
Other relatives in the home ^1^	0.637	0.545	0.712	0.649	0.913	0.602	0.090	0.309
Child’s sex ^1^	0.038 *	0.820	0.933	0.905	0.835	0.628	0.979	0.727
Child’s age (<13/>13 years) ^1^	0.443	0.507	0.224	0.169	0.042 *	0.070	0.859	0.826
Additional impairments ^1^	0.724	0.296	0.723	0.597	0.396	0.175	0.091	0.129

* *p* < 0.05. ^1^ Tau_b_. ^2^ Kruskal–Wallis.

**Table 4 ijerph-17-06644-t004:** Association between social support factors and caregiver-reported participation.

Participation Domains or Activities	Family		Spousal		Social		Professional	
	India	South Africa	India	South Africa	India	South Africa	India	South Africa
**Participation, as Reported by Caregivers**								
Overall	0.248	0.850	0.923	0.343	0.05 *^1^	0.735	0.504	0.625
Informal domain	0.165	0.846	0.902	0.549	0.123 *^,^^3^	0.462	0.681	0.960
Formal domain	0.244	0.365	0.776	0.181	0.017 *^,^^1^	0.882	0.852	0.301
Recreational activities	0.097	0.995	0.300	0.675	0.611	0.261	0.606	0.474
Active-physical activities	0.937	0.696	0.880	0.990	0.453	0.682	0.255	0.998
Social activities	0.335	0.221	0.542	0.044 *^1^	0.062	0.863	0.378	0.496
Skills-based activities	0.322	0.214	0.369	0.107	0.207	0.963	0.252	0.265
Self-improvement activities	0.357	0.651	0.251	0.431	0.377	0.484	0.908	0.677
**Participation Intensity, as Reported by Caregivers**								
Overall	0.521	0.020 *^,^^2^	0.344	0.450	0.560	0.007 **^,^^1^	0.268	0.076
Informal domain	0.803	0.042 *^,^^1^	0.874	0.450	0.860	0.040 *^,^^1^	0.056	0.172
Formal domain	0.700	0.136	0.109	0.781	0.009 **^,^^1^	0.066	0.004 **^,^^2^	0.477
Recreational activities	0.793	0.744	0.430	0.831	0.790	0.187	0.821	0.421
Active-physical activities	0.913	0.174	0.849	0.690	0.238	0.545	0.723	0.083
Social activities	0.903	0.428	0.762	0.711	0.832	0.034 *^,^^1^	0.016 *^,^^1^	0.302
Skills-based activities	0.946	0.270	0.095	0.397	0.363	0.087	0.942	0.803
Self-improvement activities	0.458	0.029 *^,^^1^	0.754	0.366	0.842	0.075	0.098	0.705
**Participation with Whom, as Reported by Caregivers**								
Overall	0.688	0.300	0.250	0.048 *^,^^1^	0.379	0.055	0.878	0.046 *^,^^1^
Informal domain	0.766	0.261	0.256	0.042 *^,^^1^	0.544	0.035 *^,^^1^	0.513	0.018 *^,^^1^
Formal domain	0.665	0.407	0.429	0.278	0.131	0.654	0.311	0.887
Recreational activities	0.908	0.535	0.033 *^,^^1^	0.302	0.637	0.008 **^,^^1^	0.765	0.058
Active-physical activities	0.423	0.298	0.453	0.066	0.530	0.543	0.511	0.983
Social activities	0.700	0.212	0.193	0.081	0.465	0.134	0.053	0.348
Skills-based activities	0.276	0.114	0.451	0.250	0.784	0.232	0.134	0.716
Self-improvement activities	0.452	0.622	0.601	0.112	0.247	0.123	0.189	0.070
**Participation Where, as Reported by Caregivers**								
Overall	0.275	0.119	0.080	0.044 *^,^^1^	0.315	0.284	0.938	0.896
Informal domain	0.262	0.107	0.036 *^,^^1^	0.034 *^,^^1^	0.832	0.448	0.733	0.854
Formal domain	0.422	0.025 *^,^^1^	0.889	0.093	0.001 **^,^^2^	0.206	0.258	0.994
Recreational activities	0.797	0.651	0.739	0.751	0.243	0.056	0.439	0.338
Active-physical activities	0.275	0.044 *	0.233	0.045 *	0.130	0.781	0.589	0.539
Social activities	0.155	0.891	0.029 *^,^^1^	0.287	0.455	0.615	0.109	0.542
Skills-based activities	0.690	0.015 *^,^^1^	0.699	0.083	0.197	0.111	1.000	0.909
Self-improvement activities	0.241	0.251	0.992	0.122	0.671	0.123	0.135	0.459
**Participation Enjoyment, as Reported by Caregivers**								
Overall	0.121	00.761	0.305	0.424	0.145	0.694	0.036 *^,^^2^	0.283
Informal domain	0.079	0.801	0.273	0.800	0.993	0.817	0.046 *^,^^1^	0.205
Formal domain	0.444	0.603	0.425	0.036 *^,^^1^	0.045 *^,^^1^	0.968	0.647	0.848
Recreational activities	0.121	0.641	0.331	0.900	0.157	0.615	0.121	0.825
Active-physical activities	0.024 *^,^^2^	0.368	0.373	0.516	0.165	0.481	0.033 *^,^^2^	0.915
Social activities	0.089	0.533	0.100	0.795	0.256	0.230	0.590	0.199
Skills-based activities	0.137	0.123	0.763	0.685	0.575	0.518	0.267	0.546
Self-improvement activities	0.877	0.870	0.198	0.104	0.725	0.212	0.070	0.116

Tau_b_ ** *p* < 0.01, * *p* < 0.05. Effect sizes: ^1^ Small effect: Tau_b_ > 0.7, ^2^ Medium effect: Tau_b_ > 0.21, ^3^ Large effect: Tau_b_ > 0.50 [79].

**Table 5 ijerph-17-06644-t005:** Significant associations between social support factors and child-reported participation.

Participation Domains and Activities	Family		Spousal		Social		Professional	
	India	South Africa	India	South Africa	India	South Africa	India	South Africa
**Participation, as Reported by Children**								
Overall		0.009 **^,^^1^		0.243		0.048 *^,^^1^		0.525
Informal domain		0.010 *^,^^1^		0.228		0.040 *^,^^1^		0.363
Active-physical activities	0.048 *^1^	0.019 *^,^^1^	0.142	0.397	0.475	0.291	0.631	0.938
Social activities		0.010 *^,^^1^		0.062		0.007 *^,^^1^		0.191
Skills-based activities		0.022 *^,^^1^		0.098		0.053		0.756
**Participation Intensity, as Reported by Children**								
Social activities	0.701		0.438		0.041 *^,^^1^		0.571	
Recreational activities		0.068		0.336		0.012*^,^^1^		0.053
**Participation with Whom, as Reported by Children**								
Recreational activities		0.033 *^,^^1^		0.539		0.758		0.066
**Participation Where, as Reported by Children**								
Formal domain	0.565		0.280		0.006 **^,^^1^	0.054		
Self-improvement activities	0.025 *^,^^1^		0.145		0.100	0.330		
Informal domain		0.014 *^,^^1^		0.416		0.202		0.928
Social activities		0.013 *^,^^1^		0.103		0.082		0.468
**Participation Enjoyment, as Reported by Children**								
Formal domain	0.049 *^,^^1^		0.216		0.301		0.072	
Active-physical activities		0.552		0.380		0.539		0.015 *^,^^1^
Self-improvement activities		0.013 *^,^^1^		0.112		0.629		0.527

Tau_b_ ** *p* < 0.01, * *p* < 0.05. Effect sizes: ^1^ Small effect: Tau_b_ > 0.7.

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
