# Peer review of "The Role of Social Support in Participation Perspectives of Caregivers of Children with Intellectual Disabilities in India and South Africa"

_ijerph, 2020, doi:10.3390/ijerph17186644_

Round 1
Reviewer 1 Report
Interesting work which shows new data on a under researched aspec of disability.
1. It will help the reader if a synthetic summary of the institutional resources available in India and SouthAfrica will be available (how and to whom schools and state institutions offer support and of what kind)
2. Page 203, 379 383 Use "sex" instead of "gender": yours is a scientific paper not a sociopsyicological one, beside for the assessment registry data were used (I suppose) so...
Author Response
Reviewer 1
Comments and Suggestions for Authors
Interesting work which shows new data on a under researched aspect of disability.
- It will help the reader if a synthetic summary of the institutional resources available in India and South Africa will be available (how and to whom schools and state institutions offer support and of what kind)
In both countries education for disabilities is provided in special schools. These schools can be government funded or private. The supports provided in these schools are dependent on a variety of factors including the context (rural or urban), the amount of funding received as well as the amount of fees paid by parents. In this study both private and public schools were represented in both India and South Africa.
- Page 203, 379 383 Use "sex" instead of "gender": yours is a scientific paper not a sociopsyicological one, beside for the assessment registry data were used (I suppose) so...
This change has been made.
Reviewer 2 Report
Dear authors,
I think it's OK for me this interesting study, but 2 data also interesting for me and they are'nt:
- Howmuch was the time during the study and how was the synchrony with the both countries?
- Do you know the type of intellectual disabilities, Down's syndrome, X fragile syndrome, unknown disability, intelectual disability for problems during delivery (anoxia), autism, and others...?
Author Response
Reviewer 2
Dear authors,
I think it's OK for me this interesting study, but 2 data also interesting for me and they are'nt:
- How much was the time during the study and how was the synchrony with the both countries?
Data was collected simultaneously in India and South Africa, over a period of two months.This has been added to the manuscript.
- Do you know the type of intellectual disabilities, Down's syndrome, X fragile syndrome, unknown disability, intellectual disability for problems during delivery (anoxia), autism, and others...?
Inclusion criteria for this study were a primary diagnosis of intellectual disability. As it has been reported that disability does not influence participation, the authors felt that it was not necessary to specify the disability resulting in the intellectual disability (Almqvist & Granlund, 2005).
Reviewer 3 Report
Introduction
- The introduction begins as if it were a discussion, without making an approach to the topic it intends to address.
- The objectives that appear at the end of the discussion, are not clearly defined, doing so in the section on materials and methods, which should not include the objectives of the study.
Material and methods
- It does not specify at any point what type of study it is, it is RCT, prospective study ...
- Subsection 2.5 of participants, which describes the characteristics of the sample, should be included in the results, since they are results.
- In the statistical analysis, no normality analysis of the sample is mentioned prior to performing a non-parametric test or if there are differences between gender or countries.
- This section, that of material and methods as a whole, should be reorganized, since it does not follow a logical order, starting by including the objectives at the end of the introduction
Author Response
Reviewer 3
Comments and Suggestions for Authors
Introduction
- The introduction begins as if it were a discussion, without making an approach to the topic it intends to address.
We attempted to clarify the approach to the topic by presenting the role of participation in development, and the knowledge and gaps in current research. Social support, the role and definitions of this in order to provide context for the readers.
- The objectives that appear at the end of the discussion, are not clearly defined, doing so in the section on materials and methods, which should not include the objectives of the study.
The objectives of the study have been moved to directly after the introduction.
Material and methods
- It does not specify at any point what type of study it is, it is RCT, prospective study ...
The study was a comparative group design with participants selected using purposive sampling. A note in this regard has been added in the revision.
- Subsection 2.5 of participants, which describes the characteristics of the sample, should be included in the results, since they are results.
According to the guidelines of this journal, the participant data has been left in the methodology section.
Thoughts?
- In the statistical analysis, no normality analysis of the sample is mentioned prior to performing a non-parametric test or if there are differences between gender or countries.
As certain data used in this article was also considered in another article in this special edition (Dada, Bastable, Schlebusch and Halder, The participation of children with intellectual disabilities: Including the voices of children and their caregivers in India and South Africa), the assessment of normality had been conducted for the previous article, and the data found to be not-normally distributed hence non-parametric tests were used. A note in this regard has been included in the text.
Differences between demographic information including sex and countries are represented in Table 1.
- This section, that of material and methods as a whole, should be reorganized, since it does not follow a logical order, starting by including the objectives at the end of the introduction
Changes have been made in the organisation of the methods and materials section in order to organise this better.
Thank you.